# Comparative Analysis of Machine Learning Models for Obstructive Sleep Apnea and Hypopnea Detection Using Pulse Oximetry and Heart Rate Variability

Vedang Sharma
*School of Computing and Augmented Intelligence*
*Arizona State University*
Tempe, USA
vshar117@asu.edu

Md Ariful Islam
*School of Electrical, Computer and Energy Engineering*
*Arizona State University*
Tempe, USA
misla122@asu.edu

Umesh Goswami
*Pulmonary Medicine*
*Mayo Clinic*
Phoenix, USA
goswami.umesh@mayo.edu

Chad M. Ruoff
*Pulmonary Medicine*
*Mayo Clinic*
Phoenix, USA
ruoff.chad@mayo.edu

Md Mobashir Hasan Shandhi
*School of Electrical, Computer and Energy Engineering*
*Biodesign Institute*
*Arizona State University*
Tempe, USA
mobashir.shandhi@asu.edu

*Abstract*—Obstructive sleep apnea and hypopnea syndrome (OSAHS) is a significantly underdiagnosed condition that can lead to dangerous and sometimes life-threatening complications such as heart failure, stroke, and sudden cardiac death. Traditional diagnostic methods for OSAHS, such as polysomnography, are resource-intensive and not readily accessible for large-scale screening. In this study, we compared the efficacy of machine learning (ML) algorithms using non-invasive physiological data-pulse oximetry and heart rate variability, which can be recorded using wearable sensors, to detect OSAHS in a large dataset consisting of 6399 recordings (53% women and mean age 62±13 years). The ML algorithms were trained and tuned using nested cross-validation on a subset of the dataset (training set, 80% of the dataset) and separately validated on the independent test set (20% of the dataset) to showcase the generalizability of our model performance. Furthermore, we investigated the performance of ML algorithms with respect to the sampling frequency, available data length, and presence of noise in physiological signals to understand the impact of real-world constraints on OSAHS detection. We also explored the model explainability with SHapley Additive exPlanations (SHAP) and an ablation study to enhance the clinical interpretation of the results. Our comparative analysis of ML algorithms (Random Forest, Support Vector Machine, eXtreme Gradient Boosting, Multi-Layer Perceptron, etc.) demonstrated the best performance for eXtreme Gradient Boosting algorithms with an F1-score of 0.896±0.012 and 0.897 on the cross-validated training set and independently validated test set, respectively. The algorithm's performance deteriorated with reduced data availability in the independent test set, with an F1-score of 0.897, 0.89, 0.887, 0.885, and 0.879 using physiological data with eight (full-night), four, two, one-hour, and 30-minute recording lengths, respectively. Algorithm performance was highest in models using pulse oximetry data with a 0.5 Hz sampling rate compared to 1 and 0.25 Hz sampling rates. The findings highlight the potential of various ML-driven analyses of unobtrusive physiological signals for scalable OSAHS screening and consideration of real-world constraints on the ML algorithm performance.

*Keywords—Obstructive sleep apnea, machine learning, pulse oximetry, heart rate variability, polysomnography*

## I. INTRODUCTION

Obstructive Sleep Apnea and Hypopnea Syndrome (OSAHS) is a common sleep disorder in which the muscles in the back of the throat relax excessively during sleep, causing the airway to narrow or close and leading to repeated pauses in breathing throughout the night [1], [2]. OSAHS represents a global public health crisis, with a 2019 analysis estimating that 936 million adults aged 30-69 are affected worldwide [3]. The disorder is prevalent and carries severe health consequences. Decades of research have established OSAHS as an independent risk factor for numerous cardiovascular and systemic diseases, e.g., heart failure, stroke, cardiac arrhythmias, sudden cardiac deaths, and hypertension [4]. The repetitive cycles of hypoxia and sympathetic nervous system activation during OSAHS act as direct drivers of pathology, with studies showing a clear dose-response relationship between OSAHS severity and the future development of hypertension [5], [6]. This link is so profound that OSAHS is now recognized as the most common identifiable secondary cause of resistant hypertension, affecting up to 83% of patients in that group [7], [8]. Beyond the clinical burden, the economic impact of undiagnosed OSAHS is staggering. In the United States alone, the annual economic cost is estimated to be approximately $149.6 billion [10]. Compounding this issue is a severe diagnostic gap; it is estimated that as many as 80% of individuals with moderate to severe OSAHS remain undiagnosed and untreated [11].

The primary bottleneck in addressing this crisis lies with the diagnostic gold standard, in-laboratory polysomnography (PSG) [2]. While diagnostically powerful, PSG is a costly, labor-intensive, and inconvenient procedure that is unsuitable for mass screening [12]. Furthermore, the unfamiliar laboratory environment can disrupt sleep (a phenomenon known as the 'first-night effect'), potentially yielding unrepresentative results [13]. Furthermore, the significant night-to-night variability of OSA means a single-night study may misclassify disease severity in a substantial percentage of patients [14]. These constraints make PSG inaccessible for much of the at-risk

population, creating the urgent need for a more scalable diagnostic paradigm.

To overcome these limitations, research has shifted towards developing automated screening tools that leverage machine learning (ML) and deep learning (DL) algorithms and a reduced set of physiological signals [15]. The feasibility of this approach is rooted in the fact that apneic events produce distinct and repeating physiological signatures in signals that are easily monitored, most notably the electrocardiogram (ECG) and peripheral oxygen saturation (SpO2) [16], [17]. These signatures include characteristic oxygen desaturations and cyclical variations in heart rate, providing the fundamental patterns that ML algorithms can be trained to recognize with high accuracy [17], [18]. Numerous studies have demonstrated the superiority of combining these two signals, as their fusion provides complementary information that enhances diagnostic confidence and robustness against noise [19], [20], [21].

While promising, many existing studies evaluate ML models

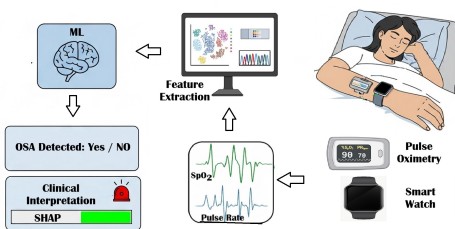

Fig. 1. Overview of a wearable system with a smart watch or pulse oximeter, capable of recording pulse oximetry and pulse (heart) rate variability information that can be leveraged to detect obstructive sleep apnea (OSAHS), which can be further incorporated with clinical interpretability.

using high-quality ECG and SpO2 signals derived directly from PSG recordings [22]. This does not fully address how these models would perform in a real-world setting using data from consumer-grade wearable devices, which may be subject to more noise and lower resolution. This study aims to bridge that critical gap.

Therefore, the primary objective of this paper is twofold: first, to conduct a rigorous comparative analysis of several machine learning models for OSAHS detection using heart rate variability (HRV) and SpO2 data from gold-standard ECG and pulse oximetry signals, respectively; and second, to systematically evaluate the robustness and performance degradation of these same models when tested on signals that have been artificially degraded to simulate the data quality of wearable monitoring devices. This approach will provide crucial insights into the real-world viability of these algorithms for developing scalable, accessible, and reliable home-based OSAHS screening technologies (Fig. 1).

## II. Methodology

This section details the framework of our study, from data acquisition and preprocessing to the experimental design for training and evaluating the machine learning models under four distinct data conditions.

### A. Dataset and Cohort Description

This study utilized data from two large, well-characterized public cohorts: the Sleep Heart Health Study (SHHS) [23] and the Cleveland Family Study (CFS) [24]. Both datasets were obtained from the National Sleep Research Resource (NSRR)

TABLE I. Demographic Distribution

|  | All (N=6399) | OSAHS Positive (N=5151) | OSAHS Negative (N=1248) |
|---|---|---|---|
| Age | 61.54 ± 12.75 | 55.06 ± 14.3 | 63.15 ± 11.82 |
| Sex |  |  |  |
| Male | 3031(47%) | 2701(52%) | 330(26%) |
| Female | 3368(53%) | 2450(48%) | 918(74%) |
| BMI | 28.72 ± 5.85 | 26.97 ± 5.39 | 29.14 ± 5.75 |
| AHI |  |  |  |
| Normal (0 − 5) | 1248 | 0 | 1248 |
| Mild (5-15) | 2387 | 2387 | 0 |
| Moderate (15-30) | 1686 | 1686 | 0 |
| Severe (>30) | 1078 | 1078 | 0 |

[25], which requires a formal data use agreement. The details of specific cohorts are described in their foundational publications, namely Quan et al. (1997) for SHHS [23] and Redline et al. (1995) for CFS [24].

The SHHS dataset included ECG signals sampled at 125 Hz and SpO2 signals at 1 Hz from 5787 participants. The CFS dataset provided ECG signals at 256 Hz and SpO2 signals at 1 Hz from 612 participants. SHHS has a subset of subjects with two PSG recordings. We only included the first PSG of the two recordings to have all the subjects with single PSG recordings across the two datasets. The demographic details of the combined datasets are provided in Table I. Apnea and hypopnea events for both cohorts were previously scored by certified technicians and provided in harmonized XML annotation files by the NSRR team.

For this study, the primary outcome was the binary classification of sleep apnea and hypopnea severity. A subject was labeled as positive for OSAHS if their Apnea-Hypopnea Index (AHI) [26] was ≥ 5 events/hour, and negative if their AHI was < 5.

### B. Experimental Data Conditions

To comprehensively evaluate model performance and robustness, five distinct versions of the dataset were created and tested independently:

- Baseline (Full-Night Data): As the primary benchmark, SpO2 and HRV features were extracted from the full duration (approximately 8-hour segments) of each subject's original, high-resolution overnight PSG recording.
- Simulation of Reduced Recording Duration: To simulate shorter monitoring periods or loss of data in real-world conditions, two separate datasets were created by extracting random, contiguous segments from each full-night recording. Features were then extracted from these shorter segments only. The four conditions were: 4-hour segments, 2-hour segments, 1-hour segments, 30-minutes segments
- Simulation of Degraded Signal Fidelity: To simulate the use of lower-fidelity sensing, two additional datasets were created by down-sampling the SpO2 signal across its full-night duration. We have not varied the resolution of the ECG signal and kept it at its original resolution, as the recommended sampling frequency of ECG for HRV feature extraction is at least 100-250 Hz [27]. This process utilized the decimate function from the SciPy library and resulted in two conditions: 0.5 Hz SpO2 with

full-resolution ECG, 0.25 Hz SpO2 with full-resolution ECG
- Simulation of Presence of Unimodal Physiological Signal: To simulate the presence of only one type of physiological signal, two separate datasets were created by including metrics from only from SpO2 signal and only HRV separately from the baseline data. This process resulted in two conditions: SpO2 only metrics, HRV only metrics
- Simulation of Presence of External Noise: To simulate the effects of external noise, two separate datasets were created by introducing distinct noise profiles to the baseline data. This process resulted in two conditions: Motion artifacts only, Additive Gaussian White Noise with Baseline Wander (AWGN-BW)

*C. Feature Engineering and Extraction*

We have extracted a total of 41 features (Table II), including three demographic and anthropometric, 23 SpO2 [28], [29], [30], and 15 HRV [31] features from the pulse oximetry and ECG signals independently for each of the experimental conditions mentioned above.

*D. Experimental Design and Model Development*

*a) Dataset Split:* To set up our classification framework, we split the cohorts into training (80%) and testing (20%) sets using a subject-wise stratified method. This stratification was based on the presence of apnea and hypopnea, along with a combination of demographic features including age, sex, and BMI. This approach ensures that no single subject contributes to both sets and that the same ratio of subjects with our outcome variable (i.e., OSAHS positive and negative) and the specified demographic factors is maintained in both the training and testing sets. We could not maintain the ratio of race and ethnicity in the two sets, as there is significant missingness and mismatches between how race and ethnicity are originally reported in the SHHS and CFS datasets.

*b) Feature Exploration with Unsupervised Learning:* Following feature extraction and dataset split, we have utilized principal component analysis (PCA) [32] and t-Distributed Stochastic Neighbor Embedding (t-SNE) [33] to reduce the dimensionality of the data and explore if we can visually see any clusters in the feature sets with respect to our outcome variable. We have only used the training set data from the baseline data (full-night recording) for this purpose.

*c) ML Model Development:* Following our feature extraction and exploration, we developed and tuned our ML models using a nested cross-validation approach to ensure robust tuning and evaluation, a best practice for clinical models [40], and separately validated the best-performing models, based on the training set results, on the independent test set. As our dataset is heavily imbalanced (80% OSAHS positive and 20% OSAHS negative), we used the Synthetic Minority Over-sampling Technique (SMOTE) [34] during ML model development to address the dataset imbalance. To compare the performance of ML algorithms in detecting OSAHS, we evaluated six algorithms, which are most commonly used in the literature [35]: Logistic Regression, Gaussian Naive Bayes (GNB), Support Vector Machine (SVM), Multi-Layer Perceptron (MLP), Random Forest (RF), and eXtreme Gradient Boost (XGBoost).

TABLE II. SpO2 AND HRV FEATURES EXTRACTED

| Feature Type | Feature Name | Number of Features |
|---|---|---|
| Demographic | Age, Sex, Body Mass Index | 3 |
| SpO2 (from Pulse Oximetry) | | |
| Statistical | Mean, Median, Standard Deviation, Minimum Value, Maximum Value, Skewness, Kurtosis | 7 |
| Clinical Indices | Delta Index, Time Below 90%, Saturation, Time Below 80%, Saturation | 3 |
| Frequency-Domain | Power in 0.014-0.033Hz band, Power in 0.017-0.050Hz band | 2 |
| Non-Linear / Entropy | Shannon Entropy, Sample Entropy, Approximate Entropy, Multiscale Entropy, Lempel-Ziv Complexity, Fuzzy Entropy, Phase Entropy, Permutation Entropy, Incremental Entropy | 11 |
| HRV (from ECG) | | |
| Statistical | Mean of R-R intervals, Median of R-R intervals, Standard Deviation of R-R intervals, Minimum of R-R intervals, Maximum R-R interval, Skewness of R-R intervals, Kurtosis of R-R intervals | 7 |
| Time-Domain | Mean of NN intervals, Standard Deviation of NN intervals, Root Mean Square of Successive Differences, Percentage of successive NN intervals > 50ms | 4 |
| Frequency-Domain | Very Low Frequency Power, Low Frequency Power, High Frequency Power, Low Frequency to High Frequency Power Ratio | 4 |
| Total | | 41 |

We used 10 outer folds for model training with five inner folds for hyperparameter tuning in the nested cross-validation on the training set. Following the training and tuning the model on the training set, we chose the best-performing model and trained this model on the full training set and separately validated it on the independent test set. As the dataset is imbalanced, we chose F1-score as the metric to maximize our model performance in the nested cross-validation, as F1-score is a widely accepted performance metric for ML classification models on imbalanced datasets [36]. We reported the performance of the cross-validated training set and test set in terms of accuracy, precision, recall (sensitivity), F1-score, specificity, area under the curve (AUC) of the receiver operating characteristic curve (ROC), and AUC of the precision-recall (PR) curve in Table III.

*d) ML Model Comparison:* Following our initial ML model development using the baseline (full night data using both SpO2 and HRV features), we chose the XGBoost algorithm, based on our initial analysis, to further explore the model performance with different feature sets, available data length, sampling frequency, and presence of noise. We have compared the performance of the XGBoost algorithms trained on datasets from a total of eleven variations: 8-hour (full night) SpO2 (1 Hz) and HRV (256/125 Hz) data- baseline data, 8-hour (full night) SpO2 (1 Hz) data, 8-hour (full night) HRV (256/125 Hz) data, 4-hour SpO2 (1 Hz) and HRV (256/125 Hz) data, 2-hour SpO2 (1 Hz) and HRV (256/125 Hz) data, 1-hour SpO2 (1 Hz) and HRV (256/125 Hz) data, 30-minute SpO2 (1 Hz) and HRV (256/125 Hz) data, 8-hour (full night) SpO2 (0.5 Hz)

data, 8-hour (full night) SpO2 (0.25 Hz) data, 8-hour (full night) SpO2 (1 Hz) and HRV (256/125 Hz) data with motion artifacts, 8-hour (full night) SpO2 (1 Hz) and HRV (256/125 Hz) data with additive white Gaussian noise with baseline wander (AWGN)

Similar to the initial model development, we used nested cross-validation to develop and tune the models on the training set data and separately validated the trained and tuned models on the independent test set data, and reported the performance of the cross-validated training set and test set in Table IV.

### E. ML Model Explainability

To comprehensively evaluate feature importance and enhance the clinical interpretation of our results, we conducted a systematic ablation study complemented by a SHapley Additive exPlanations (SHAP) [37] The ablation study utilized a leave-one-out methodology, where each of the 38 features was individually removed, and the model was re-evaluated against a baseline trained on all features using the same nested cross-validation procedure. Feature importance was primarily quantified by the change in F1-score and further characterized by the absolute performance drop, relative percentage drop, statistical significance (p-value from a paired t-test), and effect size (Cohen's d). To provide local, instance-specific insights, we further explored the best-performing XGBoost model with the SHAP TreeExplainer [38] The SHAP values were calculated on the preprocessed test set, using a background of 100 samples

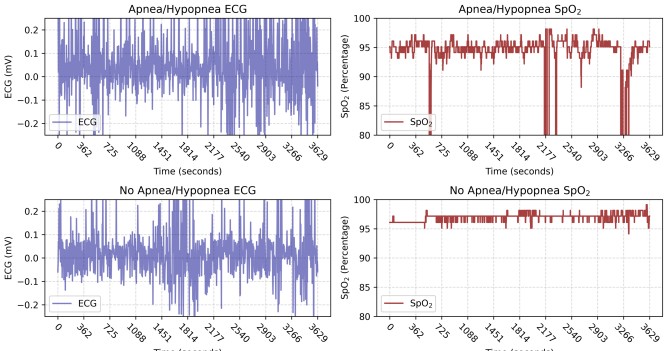

Fig. 2. Raw pulse oximetry (SpO2) and electrocardiogram (ECG) signals from two representative subjects, one with OSAHS positive and one negative.

from the training data to establish a baseline for the feature contribution analysis.

## III. RESULTS AND DISCUSSION

Fig. 2 shows raw SpO2 and ECG data from two representative subjects, one with severe OSAHS (AHI Index: 53.15) and one with OSAHS negative (AHI Index: 1.59). The physiological recordings for the apnea patient clearly demonstrate pathological patterns. The ECG signal (top left) exhibits marked variability, indicative of heart rate fluctuations often associated with respiratory disturbances. Concurrently, the SpO2 signal (top right) is characterized by recurrent, sharp desaturation events, creating a distinct cyclical pattern where oxygen levels frequently fall and recover.

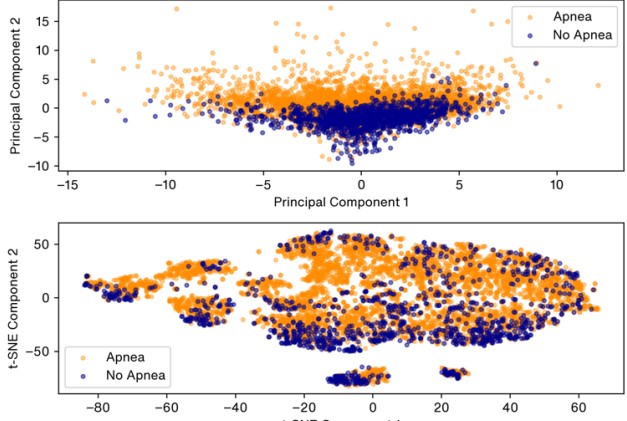

Fig. 3. Visual comparison of features from baseline (full-night) pulse oximetry and heart rate variability data with the target variable using (top) PCA and (bottom) t-SNE.

In stark contrast, the recordings from the OSAHS negative subject are distinguished by their stability. The SpO2 signal (bottom right) remains consistently high, lacking the significant desaturation events seen above. The corresponding ECG (bottom left) also shows a more regular cardiac rhythm and less motion artifact from motion or respiration. This direct comparison highlights the distinct physiological signatures that our feature-based approach is designed to quantify for automated OSAHS detection.

Fig. 3 shows the PCA and t-SNE plot. The PCA projection (top panel) reveals a significant overlap between the 'Apnea' and 'No Apnea' classes, which are concentrated in a single, dense distribution. The absence of any clear linear separability in this view highlights the need for advanced non-linear classification algorithms. In contrast, the t-SNE visualization (bottom panel) successfully captured the complex non-linear

TABLE III. MODEL PERFORMANCE METRICS ON BASELINE DATA

| Algorithms | Cross-Validated Training Set | | | | | | | Test Set | | | | | | |
|---|---|---|---|---|---|---|---|---|---|---|---|---|---|---|
| | Accuracy | Precision | Recall/Sensitivity | F1-Score | Specificity | AUC-ROC | AUC-PR | Accuracy | Precision | Recall/Sensitivity | F1-Score | Specificity | AUC-ROC | AUC-PR |
| Logistic Regression | 0.7699 ± 0.0217 | **0.9336 ± 0.0131** | 0.7690 ± 0.0269 | 0.8430 ± 0.0168 | 0.7737 ± 0.0483 | **0.8535 ± 0.0159** | **0.9586 ± 0.0054** | 0.7553 | **0.9335** | 0.7495 | 0.8314 | 0.7791 | **0.8510** | **0.9592** |
| Gaussian Naive Bayes | 0.5369 ± 0.0151 | 0.9207 ± 0.0063 | 0.4647 ± 0.0198 | 0.6174 ± 0.0178 | **0.8348 ± 0.0143** | 0.7465 ± 0.0232 | 0.9071 ± 0.0089 | 0.5457 | 0.9212 | 0.4767 | 0.6283 | **0.8313** | 0.7555 | 0.9133 |
| Support Vector Machine | 0.8041 ± 0.0061 | 0.8071 ± 0.0031 | **0.9942 ± 0.0071** | 0.8909 ± 0.0036 | 0.0200 ± 0.0195 | 0.5000 ± 0.0000 | 0.8049 ± 0.0006 | 0.8061 | 0.8074 | **0.9971** | 0.8923 | 0.0161 | 0.5 | 0.8053 |
| M | 0.8137 ± 0.0199 | 0.8942 ± 0.0134 | 0.8719 ± 0.0197 | 0.8827 ± 0.0130 | 0.5735 ± 0.0596 | 0.8401 ± 0.0217 | 0.9532 ± 0.0079 | 0.8084 | 0.8937 | 0.8650 | 0.8791 | 0.5743 | 0.8321 | 0.9544 |
| XGBoost | **0.8303 ± 0.0188** | 0.8823 ± 0.0111 | 0.9107 ± 0.0200 | **0.8962 ± 0.0120** | 0.4984 ± 0.0532 | 0.8392 ± 0.0176 | 0.9545 ± 0.0060 | **0.8311** | 0.8847 | 0.9087 | **0.8966** | 0.5100 | 0.8417 | 0.9571 |
| Multi-Layer Perceptron | 0.7957 ± 0.0143 | 0.8851 ± 0.0152 | 0.8583 ± 0.0294 | 0.8710 ± 0.0108 | 0.5375 ± 0.0821 | 0.7996 ± 0.0137 | 0.9367 ± 0.0054 | 0.7858 | 0.8743 | 0.8573 | 0.8657 | 0.4900 | 0.7991 | 0.9411 |

structures within the data, revealing several distinct clusters. Although these clusters are not entirely class-pure, they exhibit localized concentrations of each class. This indicates that while intricate patterns exist, the boundary between the two classes is highly complex and intertwined. Collectively, these plots justify the selection of a powerful gradient boosting model capable of learning the complex decision boundaries suggested by the t-SNE manifold.

### A. Comparison of Different ML Algorithms

Table III shows the comparison of different ML algorithms. The evaluation, conducted using nested cross-validation on the training set and validated on an independent test set, reveals that the tree-based ensemble methods—RF and XGBoost—yielded the most robust and accurate results.

While both models performed strongly, XGBoost demonstrated a superior balance of metrics on the test set, achieving the highest Accuracy (0.8311), F1-score (0.8966), and a compelling AUC-PR of 0.9571. In contrast, other models exhibited critical weaknesses. SVM, despite a high recall, had a near-zero specificity (0.0161) and an AUC-ROC of 0.5, indicating a complete failure to learn a discriminative boundary. GNB showed poor performance across all metrics, while Logistic Regression and the MLP were competent but ultimately inferior to the ensemble methods. Fig. 4 shows the receiver operating characteristic curve and precision-recall curve using the XGBoost model on the baseline data.

### B. Comparison of Different Feature Sets

Table IV shows the comparison of the performances of the XGBoost algorithm trained on different feature sets, data lengths, sampling frequency, and presence of noise. Analysis of the results in the table confirms the model's robustness and highlights several key findings. Performance is primarily driven by SpO2 features; the SpO2-only model (Test AUC-PR: 0.9587) performed on par with the full-feature

baseline (0.9571), whereas the HRV-only model's performance was substantially degraded (AUC-PR: 0.8895). Notably, the model also demonstrated high resilience to motion artifacts, as the configuration trained on data with motion artifacts maintained a strong performance (F1-score: 0.8974, AUC-PR: 0.9539) comparable to the clean baseline. However, the model performance deteriorated markedly with additive white Gaussian noise and baseline wandering (F1-score: 0.8792, AUC-PR: 0.9053).

Furthermore, the model maintained high efficacy with data segments. A segment of 4 hours (F1-score: 0.8899, AUC-PR:

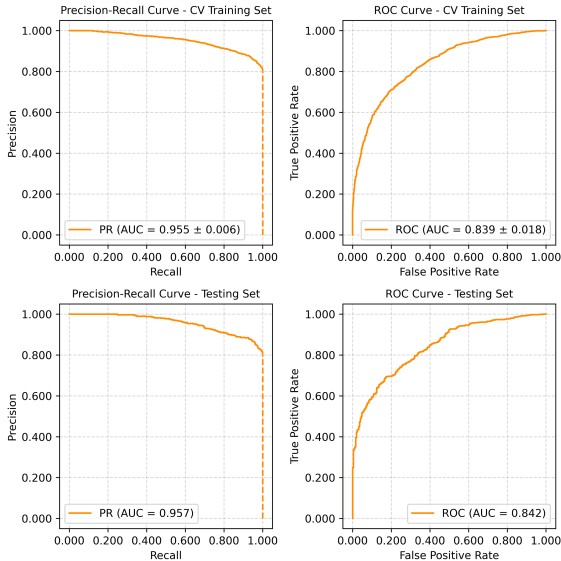

Fig.4. Precision-Recall (PR) and Receiver Operating Characteristic (ROC) curves for the optimized XGBoost model.

0.9640) yielded results similar to the full 8-hour recording. Performance remained robust even with 1-hour segments (F1-score: 0.8845, AUC-PR: 0.9469), with a more noticeable

TABLE IV. COMPARISON OF MODEL PERFORMANCE WITH DIFFERENT DATASETS, LENGTH, AND SAMPLING FREQUENCY

| Algorithms | Cross-Validated Training Set | | | | | | | Test Set | | | | | | |
|---|---|---|---|---|---|---|---|---|---|---|---|---|---|---|
| | Accuracy | Precision | Recall/ Sensitivity | F1-Score | Specificity | AUC-ROC | AUC-PR | Accuracy | Precision | Recall/ Sensitivity | F1-Score | Specificity | AUC-ROC | AUC-PR |
| Baseline Data 8-H Segments | 0.8303 ± 0.0188 | 0.8823 ± 0.0111 | 0.9107 ± 0.0200 | 0.8962 ± 0.0120 | 0.4984 ± 0.0532 | 0.8392 ± 0.0176 | 0.9545 ± 0.0060 | **0.8311** | 0.8847 | 0.9087 | 0.8966 | 0.5100 | 0.8417 | 0.9571 |
| SpO2 Features Only (1 Hz) 8-H Segments | 0.8311 ± 0.0158 | 0.8846 ± 0.0085 | 0.9088 ± 0.0180 | 0.8964 ± 0.0102 | 0.5104 ± 0.0428 | 0.8473 ± 0.0192 | 0.9570 ± 0.0064 | 0.8131 | 0.8814 | 0.8874 | 0.8844 | 0.5060 | 0.8466 | 0.9587 |
| HRV Features Only 8-H Segments | 0.7568 ± 0.0225 | 0.8500 ± 0.0102 | 0.8474 ± 0.0243 | 0.8486 ± 0.0152 | 0.3834 ± 0.0442 | 0.6850 ± 0.0188 | 0.8900 ± 0.0124 | 0.7631 | 0.8519 | 0.8544 | 0.8531 | 0.3855 | 0.6988 | 0.8895 |
| 4-H Data Segments | **0.8326 ± 0.0134** | **0.8854 ± 0.0098** | 0.9100 ± 0.0183 | **0.8974 ± 0.0089** | 0.5136 ± 0.0502 | 0.8552 ± 0.0144 | 0.9607 ± 0.0048 | 0.8210 | 0.8818 | 0.8981 | 0.8899 | 0.5020 | 0.8623 | 0.9640 |
| 2-H Data Segments | 0.8262 ± 0.0103 | 0.8782 ± 0.0085 | 0.9105 ± 0.0130 | 0.8939 ± 0.0066 | 0.4784 ± 0.0470 | 0.8303 ± 0.0211 | 0.9512 ± 0.0079 | 0.8139 | 0.8701 | 0.9039 | 0.8867 | 0.4418 | 0.8330 | 0.9548 |
| 1-H Data Segments | 0.8207 ± 0.0135 | 0.8688 ± 0.0070 | **0.9154 ± 0.0135** | 0.8915 ± 0.0085 | 0.4304 ± 0.0339 | 0.8192 ± 0.0177 | 0.9480 ± 0.0058 | 0.8088 | 0.8608 | 0.9094 | 0.8845 | 0.3936 | 0.8045 | 0.9469 |
| 30-Min Data Segments | 0.8064 ± 0.0122 | 0.8628 ± 0.0100 | 0.9032 ± 0.0149 | 0.8824 ± 0.0077 | 0.4080 ± 0.0519 | 0.7896 ± 0.0170 | 0.9376 ± 0.0052 | 0.8020 | 0.8628 | 0.8965 | 0.8793 | 0.4137 | 0.7897 | 0.9391 |
| SPO2 Features Only (0.5 Hz) 8-H Segments | 0.8217 ± 0.0155 | 0.8954 ± 0.0109 | 0.8816 ± 0.0149 | 0.8883 ± 0.0099 | 0.5745 ± 0.0500 | 0.8471 ± 0.0238 | 0.9571 ± 0.0078 | 0.8256 | 0.9063 | 0.8738 | 0.8898 | 0.6265 | 0.8669 | 0.9644 |
| SPO2 Features Only (0.25 Hz) 8-H Segments | 0.8238 ± 0.0169 | 0.9082 ± 0.0142 | 0.8692 ± 0.0109 | 0.8882 ± 0.0105 | **0.6367 ± 0.0608** | **0.8756 ± 0.0209** | **0.9678 ± 0.0061** | 0.8141 | **0.9098** | 0.8539 | 0.8810 | **0.6492** | **0.8778** | **0.9697** |
| Motion Artifacts Only 8-H Segments | 0.8249 ± 0.0124 | 0.8784 ± 0.0094 | 0.9122 ± 0.0188 | 0.8948 ± 0.0083 | 0.4347 ± 0.0589 | 0.8173 ± 0.0285 | 0.9510 ± 0.0095 | 0.8291 | 0.8847 | **0.9105** | **0.8974** | 0.4562 | 0.8156 | 0.9539 |
| AGWN-BW 8-H Segments | 0.7911 ± 0.0110 | 0.8575 ± 0.0070 | 0.8935 ± 0.0152 | 0.8751 ± 0.0070 | 0.3267 ± 0.0442 | 0.7010 ± 0.0395 | 0.9050 ± 0.0160 | 0.7970 | 0.8613 | 0.8979 | 0.8792 | 0.3286 | 0.7143 | 0.9053 |

*Only XGBoost algorithm is used for this comparison

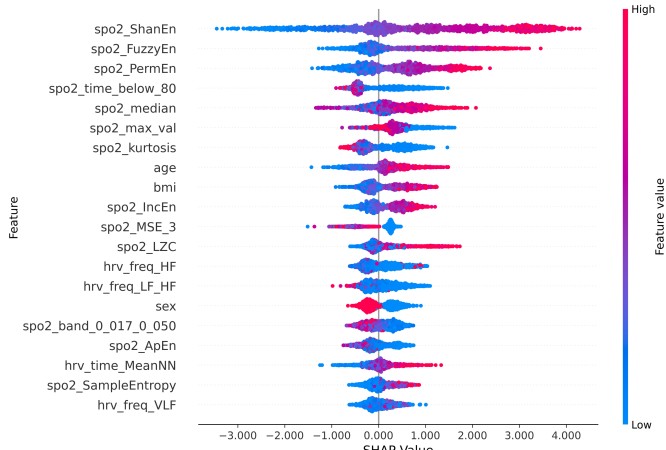

Fig. 5. SHAP Summary plot for the OSAHS Detection Model.

decline observed only when the data was reduced to 30-minute segments (F1-score: 0.8793, AUC-PR: 0.9391). This trend of graceful degradation indicates that while prolonged observation may not be essential, a minimum data duration is required for optimal ML performance.

With respect to sampling frequency, downsampling the $SpO_2$ signal to 0.5 Hz and 0.25 Hz did not impair performance but instead yielded improvements in AUC-PR, with the 0.25 Hz configuration achieving the highest test AUC-PR (0.9697). These results demonstrate the feasibility of creating accurate, computationally efficient models from less granular data, a significant advantage for deployment on resource-constrained hardware.

### C. Model Explainability

Our explainability analysis, combining a systematic ablation study with SHAP values (Fig. 5), provides a cohesive and robust understanding of the model's decision-making process. Both methods confirmed the dominant predictive power of features derived from the $SpO_2$ signal, followed by HRV and demographic features. The ablation study identified spo2_max_val as the single most critical feature, as its removal caused the most significant drop in F1-score. This quantitative finding is visually complemented by the SHAP analysis, which highlights measures of signal complexity—specifically Shannon (spo2_ShanEn), Fuzzy (spo2_FuzzyEn), and Permutation (spo2_PermEn) entropy—as highly influential.

The plot reveals clinically intuitive relationships: high feature values for these entropies, indicating greater signal irregularity, strongly push the prediction towards an 'Apnea' classification (positive SHAP values). Similarly, a greater duration of desaturation (spo2_time_below_80) and a lower median $SpO_2$ value (spo2_median) are also powerful predictors of apnea. While demographic and HRV features like age and hrv_freq_HF contribute to the model, their overall impact is secondary. This explainability analysis provides transparent insight into the model's decision-making process, verifying its reliance on physiologically relevant markers.

## IV. CONCLUSION AND FUTURE WORK

In this paper, we compared the performance of various ML models in detecting obstructive sleep apnea and hypopnea

syndrome (OSAHS). Our findings demonstrate that among the evaluated algorithms, tree-based ensemble methods, and specifically the XGBoost model, delivered the most robust and superior performance. The XGBoost classifier achieved a high F1-score on the cross-validated training set and on the independent test set, confirming its strong generalization capabilities. A key contribution of this work was the systematic evaluation of model performance under simulated real-world constraints. The analysis revealed that the model's predictive power is primarily driven by features derived from the SpO2 signal, with the SpO2-only model performing nearly as well as the full-feature model. Furthermore, the model maintained high efficacy with reduced recording durations, indicating that data from the entire night's sleep is not essential and that an optimal recording window may exist for feature creation. Perhaps most significantly, we discovered that downsampling the $SpO_2$ signal to as low as 0.25 Hz did not degrade but, in fact, improved certain performance metrics, such as the AUC-PR. Model explainability analysis using ablation study and SHAP further enhanced the clinical interpretation by confirming that predictions were driven by physiologically relevant markers of sleep apnea, such as SpO2 entropy and desaturation metrics. Collectively, these results highlight the significant potential of leveraging machine learning with signals obtainable from consumer-grade wearable devices for developing scalable, reliable, and accessible screening tools for OSAHS.

Our work, while demonstrating the strong potential of ML models for OSAHS detection, has several limitations that pave the way for future research. Firstly, although we simulated real-world conditions by manipulating data length, signal fidelity, and noise, the models were developed and validated using high-quality, lab-based PSG data. The performance of these models on data collected from personal, consumer-grade devices in an uncontrolled home environment remains to be seen. Real-world data is often corrupted by motion artifacts, poor sensor contact, and other forms of noise. Although we evaluated the model performance with the presence of artifacts and noise, a real-world scenario might not be fully represented in our simulations. Furthermore, our current approach relies on a computationally intensive feature extraction pipeline, calculating 41 distinct features. This process may not be feasible for real-time applications on resource-constrained wearable devices, which have limited processing power and battery life.

Secondly, this study was confined to the evaluation of traditional ML algorithms. While XGBoost performed exceptionally well, deep learning architectures such as Convolutional Neural Networks (CNNs), Long Short-Term Memory networks (LSTMs), or Transformers might offer superior performance. These models are adept at automatically learning relevant features and temporal dependencies directly from raw or minimally processed time-series signals. Future work will focus on developing and benchmarking such deep learning models against our current results. This approach could potentially bypass the computationally expensive feature engineering step and may prove more robust in handling the noisy, complex patterns inherent in real-world sensor data.

Finally, our dataset had limitations in its demographic representation. While we included age, sex, and BMI as features, we were unable to incorporate race and ethnicity due to significant missingness and inconsistent reporting standards

between the SHHS and CFS cohorts. Given that OSAHS prevalence and physiological manifestations can vary across different ethnic groups, the absence of this data limits our ability to assess the model's fairness and generalizability across a diverse population. Future studies should prioritize using or collecting datasets with complete and consistently reported demographic information. This will be critical not only for potentially improving model accuracy but also for ensuring that these diagnostic tools are equitable and effective for all segments of the population. Addressing these limitations will be a crucial next step in translating this research into a clinically validated and widely applicable screening technology.

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
