# OpenReview forum: "Comparative Analysis of Machine Learning Models for Obstructive Sleep Apnea and Hypopnea Detection Using Pulse Oximetry and Heart Rate Variability"
_IEEE.org/EMBS/BHI/2025/Conference — BHI 2025_

### Official Review · Reviewer_zci6 · 2025-07-01
**Comparative Analysis of Machine Learning Models for Obstructive Sleep Apnea and Hypopnea Detection Using Pulse Oximetry and Heart Rate Variability**

**Confidence:** 4
**Clarity Of Writing:** great
**Clinical Significance:** good
**Methodological Novelty:** fair
**Overall Rating:** 5
**Final Rating:** 6

**Experiments And Results:**

good

**Questions For The Authors:**

Thank you for submitting your work to BHI 2025. I have the following comments and questions:

1. The paper mentions using an 80–20 training–testing split. Could you clarify whether a separate validation set was used for hyperparameter tuning or to prevent overfitting? Including details about the validation strategy would make your approach more reproducible and transparent, especially since hyperparameter optimization can significantly impact model performance.

2. In both Table III and Table IV, the model appears to perform better on the test set than on the training set, which is unusual. This raises concerns about potential data leakage or other methodological issues. I recommend double-checking your implementation to ensure the data pipeline is properly isolated between training and testing phases.

3. My primary concern lies in the use of simulated data. While the motivation—to enable large-scale OSAHS screening—is compelling, the study currently relies on data that is synthetically degraded from high-quality sources. To strengthen the paper, I suggest either (a) evaluating model performance on real-world wearable sensor data, or (b) providing a thorough quantitative comparison between the simulated and real data to justify the realism of the synthetic inputs.

**Strengths:**

1. Addressing an important problem that has been ignored before.
2. The methodology section is well written, it clearly listed the important details, including both the datasets and the feature extraction, making the result easier to reproduce.
3. The model explainability helps enhance the clinical interpretation of the result, which shows that $SpO_{2}$ can be used as dominant predictor for OSAHS.

**Summary Of The Paper:**

Obstructive sleep apnea and hypopnea syndrome (OSAHS) may potentially lead to life-threatening complications, but it is significantly under-diagnosed due to the intensive resource needed for large-scale screening. In this study, the authors propose to use machine learning methods based on non-invasive physiologic data, i.e, pulse oximetry and hear rate variability to detect OSAHS. The proposed method is tested with several machine learning models to show its generalizability and detailed analysis is performed to show the impact on model performance with reduced data availability. Highlighting both the advantages as well as the constraints of the proposed method in real-world applications.

**Weaknesses:**

1. There are some grammar mistakes and typos in the paper, please proofread the paper (i.e Section I, "suggesting that severe OSAHS can more...").
2. The sensor data is simulated by degrading high quality data. Its hard to say how representative it is comparing to real-world wearable collected data.

---

### Official Review · Reviewer_B8Ft · 2025-07-12
**Comparative Analysis of Machine Learning Models for Obstructive Sleep Apnea and Hypopnea Detection Using Pulse Oximetry and Heart Rate Variability**

**Confidence:** 5
**Clarity Of Writing:** great
**Clinical Significance:** good
**Methodological Novelty:** fair
**Overall Rating:** 6
**Final Rating:** 6

**Experiments And Results:**

good

**Questions For The Authors:**

1. Given the SpO₂-only model’s efficacy, what drove the choice of 41 features? Did you perform ablation or automated selection to isolate the minimal effective set?
2. Why does performance drop most at 2 h and peak at 4 h? Insights into sleep architecture or statistical factors here would enhance the paper’s utility.
3. How were familial clusters (e.g., related subjects in CFS) handled to prevent train/test leakage?
4.  Do you have any pilot data from consumer devices to substantiate your down-sampling approach?

**Strengths:**

1. The proposed paper utilizes 6,399 real-life recordings using nesting cross-validation, such that it will achieve statistical power and reduce overfitting.
2. The paper overcame practical constraints by systematically evaluating shorter recordings and lower sampling rates, thereby directly addressing the limitations of consumer wearables.
3. The paper demonstrates strong discriminative ability as seen in F₁ = 0.897, and it further shows that SpO₂-only feature sets can achieve near-equivalent accuracy.
4. In the area of clinical Interpretability, the paper used  Shapley Additive Explanations (SHAP) to identify key physiological drivers such as desaturation metrics, thereby enhancing transparency for potential clinical adoption.
5. The healthcare impact of this paper demonstrates how it addresses a significant gap of up to 80% of moderate-severe OSAHS cases that remained undiagnosed and creates a low-cost screening approach.

**Summary Of The Paper:**

The paper proposes a large-scale assessment of the six machine-learning methods of detection of obstructive-sleep-apnea-hypopnea syndrome (OSAHS) using pulse oximetry (SpO2) with heart-rate-variability (HRV) signal on 6399 consecutive overnight recordings. An 80/20 partitioned nested cross-validation gives the best F1-scores (0.896 +/- 0.012 training; 0.897 test) when using XGBoost. The research collects 41 features (3 demographic, 23 SpO 2, 15 HRV) and thoroughly investigates the effect of the reduced recording durations (8 h, 4 h, 2 h) and SpO 2 sampling rates (1 Hz, 0.5 Hz, 0.25 Hz). Notably, a SpO2-only model performs as well as full-feature models, including the SHAP analysis that demonstrates a dependency on physiologically relevant markers.

**Weaknesses:**

1. In the area of simulated wearable data, down-sampling does not capture motion artifacts or sensor contact issues. There is supposed to be validation on genuine wearable recordings or perhaps simulation of noise profiles based on device specifications.
2. Due to feature-extraction overhead, using a 41-feature pipeline can be too heavy for resource-constrained devices. Applying feature‐selection or developing a lightweight subset; alternatively, this paper can explore end-to-end deep models that learn directly from raw signals.
3. Lack of a deep-learning baseline due to traditional algorithms that often underutilize temporal patterns. This paper can expand on adding a benchmark against CNNs, LSTMs, or Transformers trained on minimally processed data.
4. There is no clinical comparator in the paper; absence of a head-to-head comparison with established screening tools (e.g., STOP-BANG).
5. The paper shows only a binary threshold, and it restricts output to  AHI ≥ 5 classification. The paper can extend to multi-class severity stratification or continuous AHI prediction for better clinical guidance.

---

### Official Review · Reviewer_hbMi · 2025-07-15
**Comparative Analysis of Machine Learning Models for Obstructive Sleep Apnea and Hypopnea Detection Using Pulse Oximetry and Heart Rate Variability - Review**

**Confidence:** 5
**Clarity Of Writing:** excellent
**Clinical Significance:** excellent
**Methodological Novelty:** excellent
**Overall Rating:** 8

**Experiments And Results:**

excellent

**Questions For The Authors:**

Satisfactory.

**Strengths:**

[1] The paper demonstrates strong generalizability by showing that the XGBoost model achieved an F1-score of 0.897 on an independent test set, closely matching its cross-validated training performance, which confirms the model's robustness across test data.
[2] The paper addresses real-world applicability by testing performance under reduced data durations (2-hour and 4-hour segments) and downsampled SpO₂ signals (0.5 Hz and 0.25 Hz), finding minimal performance degradation, suggesting feasibility for consumer-grade wearable devices.
[3] The study identifies SpO₂-derived entropy features, the Shannon entropy, fuzzy entropy, and permutation entropy, as the influential predictors of OSAHS, effectively aligning with known physiological markers and enabling accurate detection using minimal, unobtrusive signal inputs.
[4] The study provides detailed insight into feature importance using SHapley Additive exPlanations (SHAP) to the XGBoost model trained on full-night data, offering clinically interpretable evidence that aligns with established pathophysiology.

**Summary Of The Paper:**

This study evaluated six machine learning algorithms including eXtreme Gradient Boosting, Random Forest, and Support Vector Machine for detecting obstructive sleep apnea and hypopnea syndrome (OSAHS) using pulse oximetry and heart rate variability data from 6,399 overnight recordings. XGBoost outperformed all other models with an F1-score of 0.896 ± 0.012 on full-night data and maintained strong performance even with reduced data durations (4-hour and 2-hour segments) and lower SpO₂ sampling rates (0.5 Hz and 0.25 Hz), demonstrating feasibility for deployment on wearable devices under real-world constraints.

**Weaknesses:**

[1]  The 41 extracted features may be too computationally intensive for real-time deployment on resource-constrained devices, such as a smartwatch.

---

### Official Review · Reviewer_ef35 · 2025-07-18

**Confidence:** 4
**Clarity Of Writing:** great
**Clinical Significance:** great
**Methodological Novelty:** poor
**Overall Rating:** 3
**Final Rating:** 6

**Experiments And Results:**

great

**Questions For The Authors:**

Have you tested your method on very small window lengths, on the order of a few minutes? If so, are you able to detect specific segments where apnea events occurred during the sleep study? Typically, OSA events last less than a minute, though some may persist slightly longer. Using smaller windows should allow you to predict apnea events, which could greatly enhance the automation of OSA annotations.

Have you tested any other degradation models, perhaps ones that are more complex and accurately reflect real-world conditions? If so, how does their impact on model performance compare to the effects seen with subsampling and data length truncation?

**Strengths:**

The paper is well-written and elaborates well on its clinical significance. The introduction, in particular, effectively outlines the problem and identifies key bottlenecks. It's also effective in suggesting the logical next steps. Table 2 provides a well-organized and comprehensive overview of the engineered feature set used for model training. It clearly categorizes features by signal type (e.g., SpO$_2$, HRV) and aids in understanding how different physiological signals contribute to model performance.

All included figures are effective in visualizing the core concepts of the study. They convey all the salient points and provide a good visual summary of the project. The use of SHAP values is highly informative, showcasing the dominant role of SpO$_2$ in sleep apnea detection, with HRV features acting as supportive inputs.

The benchmarking of machine learning algorithms is comprehensive, involving the evaluation of a broad spectrum of models. Clinically relevant metrics are reported, and various tests have been carried out. Performance comparisons are made across both full-length and truncated datasets, allowing the strengths and limitations of each model to be thoroughly characterized in different data contexts

**Summary Of The Paper:**

The authors present a comparative analysis of various machine learning models for obstructive sleep apnea–hypopnea (OSAH) detection, emphasizing the challenge of commercial deployability. Using publicly available datasets, they evaluate the performance of traditional ML models on both raw data and data modified to simulate real-world constraints. To replicate the limitations of commercial wearable sensors, they experiment with reducing the duration of recordings and also decimating the sampling rate of the pulse oximeter signal. Their benchmarking protocol is extended to these constrained datasets, which consist of truncated time series and subsampled SpO$_2$ readings. The authors found the Extreme Gradient Boosting algorithm to perform the best on 8-hour recordings. An explainability analysis found SpO$_2$ to contribute the most to the detection. Further analysis showed that 4-hour sequences outperformed the original 8-hour recordings during cross-validation, although the 8-hour recordings performed better on the withheld test set. Additionally, reducing the sampling rate of the SpO₂ signal resulted in only minimal losses in accuracy and F1 score, suggesting that prediction performance remains robust under these constraints.

**Weaknesses:**

The stated bottleneck of commercial deployability is not adequately addressed. The authors subsample SpO$_2$ to 0.5 and 0.25 Hz from the input 1 Hz signal to simulate degradations. However, the CMS60C, a commercial pulse oximeter, can already sample SpO$_2$ at 1 Hz. Typically, commercial pulse oximeter readings suffer from distortions and noise due to ambient factors and are generally not as robust as clinical-grade devices. Therefore, simple subsampling is not sufficient to capture these real-world limitations.

Also, the following paper seems to outperform the submitted paper: Levy J, Álvarez D, Del Campo F, Behar JA. Deep learning for obstructive sleep apnea diagnosis based on single channel oximetry. Nat Commun. 2023 Aug 12;14(1):4881. doi: 10.1038/s41467-023-40604-3. PMID: 37573327; PMCID: PMC10423260.

The above-mentioned paper uses the same SHHS and CFS datasets, as well as several additional datasets for its evaluation. Furthermore, the proposed model performs more advanced tasks such as Apnea-Hypopnea Index (AHI) estimation, whereas the submitted paper is limited to predicting binary labels derived by thresholding the AHI scores. From the values tabulated in Table S9 (in the supplement of Levy et al.), we can observe that their proposed model outperforms all benchmarking methods mentioned in the submitted paper.

Also, it would be beneficial to include OSA detection from other physiological signals. There is prior work in detecting apnea events using respiratory signals and respiratory effort. Adding a paragraph on the same can further improve the quality of the introduction.